# Navigating Neoplasm Risk in Inflammatory Bowel Disease and Primary Sclerosing Cholangitis

**DOI:** 10.3390/cancers17132165

**Published:** 2025-06-27

**Authors:** Demis Pitoni, Arianna Dal Buono, Roberto Gabbiadini, Vincenzo Ronca, Francesca Colapietro, Nicola Pugliese, Davide Giuseppe Ribaldone, Cristina Bezzio, Ana Lleo, Alessandro Armuzzi

**Affiliations:** 1IBD Center, Department of Gastroenterology, IRCCS Humanitas Research Hospital, Rozzano, 20089 Milan, Italy; demis.pitoni@unito.it (D.P.); arianna.dalbuono@humanitas.it (A.D.B.); roberto.gabbiadini@humanitas.it (R.G.); cristina.bezzio@hunimed.eu (C.B.); 2Department of Medical Sciences, Division of Gastroenterology, University of Torino, 10126 Torino, Italy; davidegiuseppe.ribaldone@unito.it; 3Department of Biomedical Sciences, Humanitas University, Pieve Emanuele, 20072 Milan, Italy; vincenzo.ronca@hunimed.eu (V.R.); francesca.colapietro@humanitas.it (F.C.); nicola.pugliese@humanitas.it (N.P.); ana.lleo@humanitas.it (A.L.); 4Division of Internal Medicine and Hepatology, Department of Gastroenterology, IRCCS Humanitas Research Hospital, Rozzano, 20089 Milan, Italy

**Keywords:** inflammatory bowel disease, primary sclerosing cholangitis, neoplasia risk, colorectal cancer, surveillance

## Abstract

Individuals with both inflammatory bowel disease (IBD) and primary sclerosing cholangitis (PSC) are at a markedly elevated risk for cancer, particularly colorectal cancer (CRC) and cholangiocarcinoma (CCA). While the reasons behind this increased susceptibility remain poorly understood, growing evidence points to the involvement of molecular mechanisms, including microbiota-driven inflammation, bile acid dysregulation, and immune-mediated injury. This review explores emerging insights into cancer pathogenesis and highlights recent progress in diagnostic and preventive strategies. Current recommendations include annual colonoscopy for CRC and imaging combined with CA 19-9 monitoring for CCA. New molecular and imaging tools are transforming early detection and risk management in PSC-IBD patients.

## 1. Introduction

Primary sclerosing cholangitis (PSC) is a chronic cholestatic liver disease characterized by chronic inflammation, fibrosis, and multifocal stenosis involving the intrahepatic and extrahepatic bile ducts [1]. The etiology of PSC remains largely unknown, although immunological mechanisms, genetic susceptibility, and alterations in the biliary epithelium are believed to play a key role in the pathogenesis [1]. Although the clinical course of PSC is highly variable, a substantial proportion of patients progress to end-stage liver disease, for which orthotopic liver transplantation (OLT) remains the only definitive treatment; in the absence of approved disease-modifying therapies, current management is limited to symptom control and the treatment of complications [2]. Median survival from diagnosis to death or OLT has been reported to be 14.5 years in tertiary cohorts and 21.4 years in population-based studies [3,4]. New emerging therapies, including cholangiocyte organoids (Cos) and antisense oligonucleotides (ASOs), initially developed for metabolic dysfunction-associated steatohepatitis (MASH), are showing potential for PSC [5]. SiRNA-based therapies are also currently under development [5]. PSC is strongly associated with IBD, suggesting a possible pathogenetic link between the two conditions [1,6]. The prevalence of IBD in patients with PSC varies geographically, with rates of 60–80% in the United States and Northern Europe and 30–50% in Southern Europe and Asia, a difference that may reflect an underestimation due to the failure to systematically perform colonoscopy [6]. A recent meta-analysis reported an overall prevalence of PSC in patients with IBD of 2.16%, with the highest rate observed in South America and the lowest in Southeast Asia [6]. The prevalence of PSC is higher in patients with ulcerative colitis (UC) (2.47%) than in those with Crohn’s disease (CD) (0.96%), while in patients with indeterminate forms (IBD-U), the prevalence reaches 5.01% [6]. The probability of having PSC is 1.7 times higher in patients with UC than in those with CD, while in patients with IBD-U, the probability is 3.3 times higher than in UC and 4.7 times higher than in CD [6].

PSC requires a dual clinical approach, integrating the management of progressive hepatic impairment with intensive oncologic surveillance, in view of the markedly increased risk of malignancies. Patients with PSC have a significantly elevated risk of developing hepatobiliary cancers, particularly cholangiocarcinoma (CCA), the leading cause of death in this population, with an estimated cumulative risk of 7% to 15% [3,4]. In addition, patients with PSC and concomitant IBD have an increased risk of colorectal cancer (CRC) compared with patients with IBD without PSC [1]. This review examines cancer risk in patients with IBD associated with PSC (PSC-IBD), focusing on molecular mechanisms, emerging diagnostic tools, and prevention strategies for CCA and CRC, with the aim of informing improved clinical management through recent advances in surveillance.

## 2. Materials and Methods

A comprehensive literature search was conducted using PubMed to identify relevant studies published up to March 2025. The search focused on oncogenic processes, molecular mechanisms, and recent advances in diagnostic and preventive strategies for CRC and CCA in patients with inflammatory bowel disease associated with PSC-IBD. Both human and animal studies were considered to provide a broad overview of pathogenic insights and translational developments. Keywords and MeSH terms related to “IBD”, “PSC”, “colorectal cancer”, “cholangiocarcinoma”, “molecular mechanisms”, “molecular basis”, “signaling molecules”, and “cancer prevention” were used in various combinations. Articles were screened for relevance based on titles and abstracts, and full texts were reviewed for inclusion. Reference lists of key papers were also examined to identify additional sources of significance.

## 3. Clinical Phenotype of PSC-IBD

PSC is associated with a distinctive phenotype of IBD, characterized by a higher prevalence of extensive UC and CD with colonic involvement [1,7,8]. In patients with PSC-IBD, a higher incidence of pancolitis, predominant right-sided involvement, sparing of the rectum, and backwash ileitis has been reported [8,9,10]. Similarly, CD patients affected by PSC more frequently display a colonic disease rather than an isolated ileal disease [1,8].

The effect of IBD on the natural history of PSC remains unclear [1]. Emerging data report less severity of liver fibrosis and reduced spleen size in patients with active PSC-IBD compared with PSC alone, suggesting a possible protective effect of intestinal inflammation on PSC progression [11]. Lipopolysaccharide-triggered activation of hepatocellular NF-κB may inhibit bile acid metabolism, potentially explaining the observed protective effect of intestinal inflammation in these patients [11,12]. These data support the hypothesis that intestinal inflammation may modulate the progression of PSC in a protective manner [11]. Recent findings suggest that IBD, when associated with PSC, may present a more active clinical course than previously thought [13]. A higher incidence of colectomies has been observed in patients with PSC-IBD than in IBD alone, often due to refractory colitis rather than neoplasm [13]. PSC-CD patients show longer event-free survival (cancer, OLT, or death) compared to PSC-UC, possibly due to a higher prevalence of small duct PSC in CD [6,14]. As concerns pouchitis, this is more common in patients with PSC-UC than those with UC alone after proctocolectomy surgery with ileo-pouch anal anastomosis (IPAA), with a reported incidence of 63% versus 32% [15]. In addition, pouchitis in PSC tends to become chronic more frequently (68% vs. 34%, *p* < 0.001) and to respond less to antibiotics [16].

Although it has been hypothesized that therapies for IBD may influence PSC, recent meta-analyses have shown that Infliximab, Adalimumab, and Vedolizumab do not improve markers of cholestasis [17]. Vedolizumab has been associated with a slight increase in alkaline phosphatase (ALP) and other liver enzymes, with no effect on total bilirubin, but no evidence of histologic or radiologic liver progression [17]. Infliximab moderately increases total bilirubin, while Adalimumab has been associated with an isolated reduction in ALP, suggesting a possible effect on bone homeostasis [17].

## 4. Pathophysiological Mechanisms of PSC and Impact on Cancer Development

### 4.1. Intestinal Permeability and PSC

The pathophysiologic link between PSC and IBD remains complex and not fully elucidated, but several molecular and immunologic mechanisms appear to contribute to hepatobiliary damage and increased risk of malignancies, particularly CCA and CRC, in patients with PSC-IBD compared with patients with IBD alone [1]. Among these mechanisms, a central role has been attributed to increased intestinal permeability, commonly referred to as the “leaky gut” hypothesis (Figure 1) [1,18,19]. This hypothesis posits that the disruption of the intestinal epithelial barrier—due to chronic inflammation, infection, or chemical injury—facilitates the translocation of bacteria and their products, such as pathogen-associated molecular patterns (PAMPs), from the intestinal lumen into the portal circulation [1,19]. Once these microbial components reach the liver, they activate resident immune cells, particularly Kupffer cells and hepatic stellate cells, triggering an exaggerated inflammatory response characterized by the production of proinflammatory cytokines and chemokines, including tumor necrosis factor-alpha (TNF-α) [19,20]. This inflammatory cascade contributes to chronic biliary tract inflammation, portal fibrosis, and ultimately the development of PSC [1,19,20]. Experimental evidence supporting the leaky gut hypothesis comes from an animal model developed by Lichtman and colleagues [21]. In this study, bacterial overgrowth in a rat model, achieved through the creation of a surgically induced blind loop in the jejunum, led to increased bacterial translocation and resulted in cholangiographic changes resembling those seen in PSC [19,21]. These findings provide proof of concept that increased microbial translocation can directly induce biliary injury and fibrosis [19,20,21]. Clinical data further support the link between bacterial translocation and disease progression in PSC [19,22]. In a cohort study involving 166 patients with PSC and 100 healthy controls, serum levels of soluble CD14 and lipopolysaccharide-binding protein (LBP)—two classical markers of bacterial translocation—were significantly elevated in PSC patients compared to controls [22]. Importantly, higher LBP levels were associated with reduced transplant-free survival, suggesting that continuous microbial leakage from the gut may have a direct and detrimental impact on clinical outcomes in PSC [19,22]. However, not all studies have consistently demonstrated increased intestinal permeability in PSC patients [19,23]. For instance, Björnsson and colleagues evaluated gut permeability by measuring the differential urinary excretion of lactulose and L-rhamnose, a standard test for epithelial barrier integrity, in a small cohort of 22 PSC patients and found no significant difference compared to healthy controls [23]. Nevertheless, the limited sample size of this study may have precluded definitive conclusions [19]. Given the potential role of gut-derived microbial products in driving liver inflammation, therapeutic strategies aimed at modulating the intestinal microbiota have been explored [24]. The use of antibiotics such as vancomycin and metronidazole in PSC patients has demonstrated a significant reduction in ALP levels, an important biomarker of cholestasis, suggesting that microbiome-targeted therapies may have a beneficial effect by reducing microbial translocation and dampening hepatic inflammation [24].

### 4.2. Gut Microbioma and PSC

Another central mechanism is intestinal dysbiosis, which is more pronounced in patients with PSC-IBD than in PSC or IBD alone [25]. In patients with PSC, reduced alpha-diversity and increased proinflammatory bacteria, such as *Veillonella*, *Enterococcus*, *Streptococcus*, and *Lactobacillus*, have been observed, regardless of the presence of IBD (Figure 1) [26]. An association between *Enterococcus* and elevated serum ALP levels, a marker of disease severity, has been observed [27]. The genus *Veillonella* has frequently been found in excess in patients with PSC [19,28]. *Veillonella* produces primary amines that can serve as substrates for vascular adhesion protein-1 (VAP-1), promoting recruitment of effector cells to the liver [28]. This mechanism may promote aberrant lymphocyte trafficking between the gut and liver and contribute to the development of inflammation and fibrosis in PSC [19,28]. The biliary microbiota has also been observed to have unique characteristics compared with the intestinal microbiota, with enrichments of *Staphylococcus* and *Streptococcus sanguinis* in patients undergoing endoscopic retrograde cholangiopancreatography (ERCP) [29]. The mycobiome was also found to be altered in PSC-IBD patients: an increase in fungal biodiversity and proportion of *Exophiala*, an opportunistic fungus associated with infections in immunocompromised subjects, has been documented, along with a reduction in *Saccharomyces cerevisiae*, known for its anti-inflammatory effect [19,30]. In more recent studies, a connection has emerged between alterations in mucosal microbiota, gene expression, and specific immunologic signatures in PSC-IBD patients compared with patients with ulcerative colitis alone [31]. Alterations were observed in bacteria involved in bile acid 7α-dehydroxylation, an important step in the conversion of primary to secondary bile acids in the intestine, including *Escherichia*, *Megasphaera*, and *Lachnospiraceae* [19,25]. A reduction in butyrate-producing bacteria, such as *Roseburia* and *Prevotella*, was also noted [19,25]. Loss of butyrate may impair intestinal barrier integrity and disrupt immune regulation, potentially promoting disease progression [19]. Furthermore, an increase in bacteria such as *Sphingomonas*, associated with abnormal lymphocyte recruitment from the gastrointestinal tract to the liver, suggests direct mechanisms linking the gut and liver in PSC pathogenesis [19,25]. Other analyses of ileal and colonic terminal mucosa showed an increase in *Barnesiellaceae*, *Blautia*, and *Clostridiales*, bacteria known to influence regulatory T-cell differentiation and pro-inflammatory cytokine production [32,33]. However, other investigations have reported a reduction in *Clostridiales* associated with low bacterial diversity and a concomitant reduced level of fecal secondary bile acids, similar to what has been observed in patients with advanced liver cirrhosis [34,35,36]. Fecal microbiota transplantation (FMT) from a PSC patient into mouse models induced hepatobiliary damage through the translocation of *Klebsiella pneumoniae* and *Proteus mirabilis* into mesenteric lymph nodes, suggesting a possible causative role of dysbiosis in the etiopathogenesis of PSC [37]. In a pilot study of FMT in patients with PSC-IBD, a single treatment via colonoscopy increased microbial diversity in all participants and reduced by >50% ALP levels in 3 of 10 patients [38].

In summary, current studies highlight distinct alterations of the gut microbiota in PSC, showing differences not only from healthy controls but also from patients with IBD without liver disease; however, whether these changes are a cause or a consequence of the disease remains unclear [19]. While reduced microbial diversity may reflect secondary effects of chronic inflammation, the enrichment of specific taxa such as *Veillonella*—implicated in fibrosis and liver disease—suggests a possible pathogenic role [39]. Further research is needed to clarify the contribution of individual microorganisms to PSC development [19].

### 4.3. Bile Acid Metabolism and Lymphocyte Trafficking

A characteristic aspect of PSC is the dysfunction of bile acid metabolism [1,40]. In affected patients, an increased ratio of primary to secondary bile acids has been found, indicating a defect in their intestinal conversion [41]. Multi-omics studies have shown that in patients with PSC-IBD, compared with those with isolated UC, there is impaired conversion of secondary bile acids in the intestine, resulting in altered regulation of hepatic bile acid synthesis [31]. This imbalance has been correlated with increased disease activity and a more intense hepatic inflammatory response [1]. In this context, treatment with obeticolic acid, a farnesoid X receptor (FXR) agonist, showed a significant reduction in ALP levels in patients with PSC, suggesting that modulation of biliary metabolism may be a promising therapeutic target in PSC-IBD [42]. Altered biliary composition may directly increase carcinogenic risk. Bile acids can induce DNA damage, promote mutations, and favor the survival of apoptosis-resistant clones—all hallmarks of oncogenesis [43,44]. This hypothesis was supported by a prospective study of 20 patients with PSC-IBD and 25 patients with UC without PSC, which found a higher frequency of aneuploidy in the PSC-IBD group (60% vs. 20%; *p* ≤ 0.007), suggesting greater genomic instability in the colic epithelial cells of these patients [45].

Finally, aberrant trafficking of lymphocytes between the intestine and the liver plays a crucial role [46]. In patients with PSC-IBD, T cells express high levels of integrin α4β7, which is responsible for the recruitment of lymphocytes to the gut through binding to MAdCAM-1 (mucosal addressin cellular adhesion molecule 1) [46]. Studies of liver samples from patients with PSC-IBD have shown increased expression of CCR9 and CD3 in the portal tract compared with patients with PSC alone, indicating an infiltration of activated T lymphocytes from the intestine [47]. Intestinal *Firmicutes*, such as *Lachnospiraceae*, produce primary amines that act as substrates for VAP-1, promoting the expression of MAdCAM-1 in the hepatic endothelium and facilitating the infiltration of memory T cells into the bile ducts [48]. However, clinical studies with VAP-1 inhibitors and vedolizumab, which block α4β7, have not shown a significant impact on liver disease progression, suggesting the presence of compensatory mechanisms in lymphocyte trafficking in PSC-IBD [49].

### 4.4. Genetic and HLA Contributions to PSC

Genetic susceptibility in PSC has features common to both autoimmune diseases and IBD [50]. A distinguishing feature of PSC is a strong linkage to the major histocompatibility complex (HLA), similar to what is observed in classic autoimmune diseases [50]. However, it remains unclear how specific HLA variants increase the risk of developing PSC [50]. As with many other HLA-associated diseases (with the exception of celiac disease), the mechanism by which these genes influence the disease remains unknown [50]. The mechanisms that explain the role of other, non-HLA-related risk genes also remain largely hypothetical [51]. A big step forward in understanding the genetics of PSC has come thanks to a study conducted by the international PSC study group, which analyzed nearly 130,000 genetic variants in approximately 3789 patients and more than 25,079 healthy controls [52]. The study used an “immunochip,” an instrument designed to study variants already known in other autoimmune diseases [52]. This work identified 9 new genes associated with PSC and confirmed the involvement of 33 other genes already known in diseases such as CD, UC, celiac disease, psoriasis, and rheumatoid arthritis [50,52]. Among the most important genes confirmed are those located in the 2q35 (which includes TGR5 and receptors for interleukin-8, IL8RA, and IL8RB) and 19q13 (which contains the *FUT2* gene) regions [50,52].

Functionally, one of the novel biological pathways suggested by genetic associations involves the molecular network connecting *PRKD2*, *HDAC7*, and *SIK2*, although it remains uncertain whether these alterations predominantly affect T-cell activation, bile acid homeostasis, or other pathological processes [50].

Besides non-HLA genes, the genetic area most strongly linked to PSC remains the HLA [53]. However, studying this region is complicated because many genes are inherited together as building blocks (a phenomenon called “linkage disequilibrium”), making it difficult to determine which variant really plays a role in the disease [53]. One haplotype in particular, HLA-DRB1*13:01-DQA1*01:03-DQB1*06:03, has been identified as a major risk factor [54,55]. In Scandinavian populations, these genes are almost always found together, suggesting a strong connection between them [53]. However, studies in more genetically diverse populations have shown that this association is not always so close: only 65% of individuals with HLA-DRB1*13:01 also had DQB1*06:03 [53]. This finding suggests that studying genetically heterogeneous populations could help separate true causative genes from those that are merely associated, improving understanding of the mechanisms underlying PSC [53].

Despite these important findings, the interpretation of genetic data in PSC still requires caution. Indeed, the disease is very complex, and no single gene alone can explain its development.

### 4.5. Oncogenic Implications

In patients with PSC-IBD, compared with those with UC, significant downregulation of genes and pathways involved in cancer regulation is observed [31]. In particular, DNA damage-repair mechanisms, cellular checkpoints, *p53* signaling, and mitotic transition processes, such as *APC*/CCdc20-mediated cyclin A degradation, are impaired [31]. Some genes, such as *TUBB2A*, appear to play a central role in these molecular alterations [31]. Methylation of a component of the *TUBB6* gene has been shown to be significantly positively correlated with colonic dysplasia [56]. In parallel, there is a marked reduction in the expression of *PRAC1*, a gene normally associated with decreased susceptibility to right colon cancers [57]. Taken together, these changes suggest an impairment of cellular surveillance and protection mechanisms against neoplastic transformation [31]. Further observations indicate that certain beta-tubulins, such as *TUBB5*, may be involved in the altered immune response characteristic of hepatic autoimmune disorders, suggesting a potential link between immunity, alterations in the gut microbiota, and cancer risk [58]. Profound changes are also present at the biliary level, with increased abundance of microorganisms such as *Lactobacillus*, *Actinomyces*, *Alloscardovia*, and *Peptostreptococcaceae* [59]. Some of these microbial changes are associated with changes in bile acid composition, such as increased tauroursodesoxycholic acid (TUDCA), which may contribute to the promotion of an environment conducive to tumorigenesis [29]. Chronic epithelial damage, induced by multifocal biliary stenosis, the accumulation of toxic bile acids, and microbial translocation, results in the activation of cholangiocytes, leading to cell proliferation and the production of cytokines and growth factors [19]. However, it remains unclear if activated cholangiocytes directly activate B-cell antibody production and T-cell expansion [60]. This process promotes the development of a pro-oncogenic microenvironment, characterized by epithelial regeneration, fibrosis, angiogenesis, and immunoevasion [19]. In addition, activation of the peribiliary glands can lead to the formation of dysplastic lesions, which are considered precursors to CCA [61]. A higher frequency of metaplasia and dysplasia of cholangiocytes has been observed in patients with PSC-CCA, compared with those with PSC without neoplasia, regardless of the duration of the disease [60,62,63].

Collectively, these mechanisms may help explain the elevated cancer risk observed in PSC-IBD patients, though further research is needed to establish causality. The combination of chronic hepatobiliary damage, persistent inflammatory stimulation, and bile acid-induced genomic instability could contribute to progression to CCA and CRC, highlighting the need for targeted surveillance and treatment strategies.

## 5. Oncological Risk in PSC-IBD

Patients with PSC-IBD have a significantly higher risk of developing malignancies, particularly CRC and CCA, compared to patients with isolated IBD, highlighting a distinct clinical profile requiring specific management [64]. The risk of CRC in patients with PSC-IBD is significantly increased, estimated to be four to seven times higher compared to patients with IBD without PSC, with a higher prevalence of dysplasia and predominant localization in the right colon, suggesting different pathogenetic mechanisms than in isolated IBD [65]. CCA is also more frequent in patients with PSC-IBD than in PSC alone, with the incidence increasing with age at PSC diagnosis [66]. In addition, the risk of other malignancies, such as pancreatic carcinoma and gallbladder carcinoma (GBC), is significantly higher in patients with PSC-IBD compared with IBD alone, reflecting a complex interaction between the hepatobiliary system and chronic intestinal inflammation [67]. These substantial differences in the oncologic risk profile underscore the need for dedicated surveillance strategies and a personalized approach for early diagnosis and timely management in patients with PSC-IBD [67].

### 5.1. Cholangiocarcinoma

CCA represents the neoplasm most frequently associated with PSC and is a major cause of morbidity and mortality in this population [1,4,13,67,68]. In patients with PSC, the risk of developing CCA is markedly increased compared with the general population, with an estimated annual incidence of from 0.5% to 1.5%, and a cumulative incidence of up to 20% [3,67,69,70]. In a recent international cohort, including 7121 patients from 37 countries, a global prevalence of hepatobiliary malignancies of 10%, with CCA as the most common form, was reported [4]. In relative terms, the risk of CCA in patients with PSC is increased 400- to 1500-fold compared with the general population [3,71]. This risk appears further amplified in patients with PSC-IBD, particularly in the presence of UC [4,72]. In this subpopulation, the risk of CCA is significantly higher not only compared with the general population but also compared with patients with isolated PSC [1,4,13,72]. In a Swedish cohort study, biliary tract cancer was diagnosed in 234 patients with PSC-IBD, compared with only two cases expected in the general population, with a standardized incidence ratio (SIR) of 140 (95% CI: 123–159), a value approximately 56 times higher than the risk observed in patients with IBD uncomplicated by PSC [73]. In the largest available population study (N = 2588), patients with PSC and IBD were observed to have a 28-fold higher risk of developing CCA than patients with IBD in the absence of PSC [13]. Although definitive predictive biomarkers for CCA in the context of PSC remain elusive, several clinical parameters have been implicated in risk stratification. Notably, prolonged disease duration of concomitant IBD and a prior history of colonic dysplasia have emerged as significant risk modifiers that exacerbate the probability of malignant transformation in PSC-IBD cohorts [68].

Epidemiologically, the cumulative incidence of CCA among patients with PSC-IBD has been reported at 6.3% [13]. In a meta-analysis that included 16 studies with a total of 9074 patients, the aggregate incidence of CCA in subjects with PSC-IBD was 7.16 cases per 1000 person-years (95% CI: 4.48–11.44; *p* < 0.01), with high heterogeneity among studies (I² = 96%) [72]. Despite evidence of the role of PSC and IBD in promoting the occurrence of intra-hepatic (IH-CCA) and extra-hepatic (EH-CCA) cholangiocarcinoma, there is a lack of conclusive data on the extent to which their association (PSC-IBD) increases risk compared with the isolated presence of either condition [74]. Furthermore, although several studies have assessed the cumulative risk of CCA in patients with PSC, few have separately analyzed the specific risk of developing intrahepatic and extrahepatic forms [74]. A notable sex-based disparity was observed in the median age at CCA diagnosis, which occurred earlier in males (64 years; IQR 48–71) compared to females (67 years; IQR 56–79), a difference that reached statistical significance (*p* < 0.001), suggesting potential gender-based differences in disease biology or detection. Age at PSC diagnosis was positively correlated with CCA incidence, highlighting a clear age-dependent risk trajectory. Specifically, the incidence per 1000 patient-years increased from 2.4 (95% CI, 1.0–4.7) in individuals aged 18–30 years to 8.1 (95% CI, 4.5–13.3) in the 31–40 year-old age group, 11.3 (95% CI, 7.6–16.1) among those aged 41–50, and 11.9 (95% CI, 7.6–76.6) in those in the 51–60 age group, peaking at 20.2 (95% CI, 16.5–25.0) in patients over 60 years of age [13].

Multivariate analysis identified advanced age at PSC diagnosis as the sole statistically significant independent risk factor for CCA development in the PSC-IBD population, with a hazard ratio (HR) of 1.26 per 10-year increase (95% CI, 1.16–1.37; *p* < 0.001) [13].

Clinically, CCA in the context of PSC-IBD is associated with a particularly aggressive course. Among affected individuals, 84% (138 of 164 patients) succumbed to the malignancy, with a median survival time of only 0.45 years post-diagnosis (IQR 0.15–1.30), reflecting the fulminant nature of disease progression once diagnosed [13]. These findings underscore the critical importance of implementing robust surveillance protocols aimed at facilitating early detection and timely therapeutic intervention. Furthermore, due to the limited number of CCA cases among patients with PSC associated with CD or IBD-U, subgroup-specific risk analyses were not feasible, emphasizing the need for larger, targeted studies to delineate risk profiles within these subpopulations [13].

### 5.2. Colorectal Cancer

#### 5.2.1. Epidemiology and Risk Factors of CRC in PSC-IBD

In patients with PSC-IBD, a significantly increased risk of CRC has been demonstrated [13]. The 10-year risk of CRC in patients with PSC-IBD is estimated to be between 7% and 14%, being approximately 10 times higher than in the general population and 4 times higher than in patients with IBD without PSC [75,76,77]. It has also been reported that the risk of CRC in patients with PSC-IBD is increased about 6.9-fold compared with patients with IBD alone [78]. In patients with UC associated with PSC, the prevalence of dysplasia and CRC was significantly higher than in patients with isolated UC [45,79]. In contrast, in patients with CD associated with PSC, the risk of dysplasia and CRC was not significantly increased (16.3% vs. 13.8%; *p* = 0.98) [80]. A meta-analysis including 16 studies with a total of 13,379 patients showed a 3.2- and 3.4-fold increased risk of dysplasia and CRC, respectively, in patients with PSC-IBD compared with patients with IBD alone, with a significant association only in patients with UC [77]. In addition, a higher incidence of right colon cancer was found in patients with PSC-IBD [79,80].

In patients with PSC-IBD, the risk of CRC was particularly high in those diagnosed with IBD before the age of 50 years [13]. The median age at diagnosis of CRC in patients with PSC-IBD was significantly lower than in those with isolated IBD (59 years; IQR 47–72 vs. 69 years; IQR 58–78; *p* < 0.001) [13]. In patients diagnosed with IBD before age 50 years, the risk of CRC was approximately five-fold higher in patients with PSC-IBD compared with IBD alone, whereas this difference was reduced to less than two-fold in those diagnosed with IBD later in life [13]. No significant differences were found in the incidence of CRC between PSC-IBD and isolated IBD in patients diagnosed over 60 years of age [13]. Overall, the presence of PSC was associated with a more than three-fold increased risk of CRC compared with IBD alone, regardless of sex, age, race, and comorbidities [13]. This increased risk was also evident in the subanalysis of patients with UC and IBD-U, whereas in patients with CD associated with PSC, only an increasing risk trend was observed, but without reaching statistical significance [13].

In a cohort of patients with PSC-IBD, 4% of patients developed CRC. Among them, about 40% developed CRC after the diagnosis of IBD but before the diagnosis of PSC, while 30% had this event within 2 years after the onset of PSC [13]. However, the temporal sequence between IBD and PSC diagnoses was not consistently reported across the cohort, and caution is warranted in assuming a uniform disease chronology. Among patients who developed CRC later, 20% died of CRC-related causes. It was also observed that in patients who underwent annual colonoscopy, the rate of CRC-related deaths was lower than in patients who did not undergo regular monitoring (18% vs. 21%), although this difference did not reach statistical significance [13].

A study of patients with PSC and PSC-IBD evaluated the incidence of CRC over a follow-up period of approximately 37 years [81]. The cumulative incidence of CRC in patients with PSC-IBD was 4.3%, 20.4%, and 38.1% at 5, 10, and 20 years, respectively, significantly higher than in patients with PSC alone (0.0%, 13.0%, and 13.0% at 5, 10, and 20 years, respectively) (log-rank *p* < 0.01) [81]. A Kaplan–Meier analysis showed that the incidence of CRC in patients with PSC-IBD was significantly higher than that in patients with PSC alone (adjHR 21.52; 95% CI, 3.11–148.76) [81]. The relative incidence rates (IRR) of CRC between PSC-IBD and PSC alone were significantly elevated within 10 years (IRR 11.1; 95% CI, 2.0–112.2), 20 years (IRR 13.5; 95% CI, 2.6–133.2), and throughout the follow-up period (IRR 13.7; 95% CI, 2.6–135.5) [81]. Compared with the general population, the estimated risk of CRC was significantly higher both in patients with PSC alone (SIR 9.2; 95% CI, 1.1–33.2) and in patients with PSC-IBD (SIR 36.4; 95% CI, 16.6–69.1) [81]. Although current guidelines do not support an increased risk of CRC in patients with isolated PSC, a diagnostic colonoscopy can be considered every 5 years in individuals without IBD or earlier if symptoms suggestive of IBD arise [82]. Nonetheless, some studies have hypothesized that chronic exposure of the colonic mucosa to toxic secondary bile acids, such as deoxycholic acid (DCA), may promote oxidative DNA damage and a pro-neoplastic microenvironment, potentially contributing to CRC development even in the absence of clinically evident IBD [83].

Even in patients with PSC-IBD undergoing OLT, the risk of CRC remains high [79]. The 10-year cumulative incidence of CRC in patients undergoing OLT for PSC was 8.2%, compared with 2.6% in patients transplanted for other causes, although only a minority of patients transplanted for PSC had a concomitant diagnosis of IBD [84]. In addition, an incidence rate of 5.3% was reported in patients with UC after OLT for PSC compared with 0.6% in patients transplanted for non-PSC causes [85]. It has been estimated that the risk of CRC in patients with PSC-IBD undergoing OLT is four times higher than in patients without PSC. In patients with PSC-IBD undergoing OLT, CRC predominantly localizes to the right colon (cecum and ascending colon) [86,87].

In a histological study of specimens from 16 patients with PSC and IPAA, a higher proportion of atrophy and a trend toward dysplasia were found compared with controls with IPAA for IBD without PSC [88]. However, a retrospective review of 21 patients with PSC and IPAA showed no increased risk of neoplasia compared with patients with UC and IPAA without PSC [89]. To date, studies conducted on larger cohorts of patients with PSC and IPAA, including a meta-analysis of 11 studies published in 2021, have not specifically focused on the risk of pouch neoplasia as the primary outcome [90].

Regarding malignancy in the pouch after OLT, an increased risk of dysplasia and pouch carcinoma was observed in patients with PSC-IBD who underwent colectomy and OLT, compared with IBD patients without PSC who underwent colectomy alone [79,91]. A prevalence of pouchitis of 98.4% was reported in patients undergoing OLT for PSC, with a cumulative incidence of pouch dysplasia of 5.6% at 5 years [92]. The risk of malignancy in the pouch is hypothesized to be related to secondary bile acid exposure [91].

#### 5.2.2. Molecular Characteristics of CRC Associated with PSC-IBD

Patients with PSC associated with IBD have an increased risk of CRC compared with patients with IBD without PSC, but it is unclear whether the molecular phenotype of these tumors is distinct. A study conducted by de Krijger et al. characterized PSC-IBD-associated CRC (PSC-IBD-CRC), showing that changes in DNA copy number are similar to those observed in IBD-associated CRC (IBD-CRC) and sporadic CRC (s-CRC). However, mutation frequencies in PSC-IBD-CRCs are lower than in s-CRCs, while they are comparable in IBD-CRCs [93].

The most frequent mutations in PSC-IBD-CRCs involve *TP53* (63–89%), an early event in IBD-associated tumorigenesis, while mutations in *APC* are rare (5%), although frequent loss of the 5q arm is observed, suggesting alteration of the WNT pathway through copy loss rather than through direct mutations [93,94,95,96]. Only one PSC-IBD-CRC neoplasm showed microsatellite instability (MSI), confirming that this feature is rare in IBD-associated CRCs [93].

A relevant difference between PSC-IBD-CRC and IBD-CRC concerns the CIMP-positive phenotype, indicative of hypermethylation: 44% of PSC-IBD-CRCs were CIMP-positive, compared with 90% of IBD-CRCs and 34% of s-CRCs [93]. This suggests that epigenetic alterations play a role in PSC-IBD-CRCs, although less pronounced than in IBD-CRCs. Despite this difference, neither mutations in *BRAF* nor those in *KRAS* are found to be frequent in PSC-IBD-CRCs, suggesting alternative mechanisms of tumorigenesis [93].

Mutations, mainly in *TP53* and *FBXW7*, were observed in the adjacent non-dysplastic mucosa to PSC-IBD-CRCs, but the number of alterations was generally limited (zero to two per patient, with the exception of six mutations). Sixty percent of adjacent non-dysplastic mucosa samples in PSC-IBD-CRCs showed a CIMP-positive phenotype, although in some cases, the normal mucosa was CIMP-positive while the associated tumor was not, suggesting a heterogeneous distribution of methylation [93].

The study has some limitations, including the small sample size, which may have limited the statistical power to detect significant differences. The young age at diagnosis of IBD in PSC-IBD patients (79% before age 40) confirms that this is a high-risk population. The absence of obvious phenotypic differences in PSC-IBD-CRCs compared with IBD-CRCs suggests the need for further studies, through exome sequencing or epigenetic analysis, to elucidate the mechanisms of carcinogenesis and identify specific biomarkers useful for targeted surveillance strategies [93]. The distinct clinical and molecular features of CRC arising in patients with PSC-IBD compared to sporadic CRC are summarized in Table 1.

#### 5.2.3. Histopathological Characteristics of Dysplasia Associated with PSC-IBD

Retrospective studies have shown that the prevalence of dysplasia in patients with PSC-IBD is higher than in patients with IBD without PSC, with an incidence of advanced neoplasia of 37% in the PSC-IBD group compared with 22% in the IBD group without PSC (*p* = 0.035) [40]. In addition, the risk of developing dysplasia was reported as high as 45% in patients with PSC-IBD compared with 16% in patients with IBD alone (*p* ≤ 0.002) [45].

From the clinic-pathological point of view, dysplastic lesions in patients with PSC-IBD have distinctive features compared with those in patients with IBD alone [40]. Dysplasia in patients with PSC-IBD is more frequently unconventional (61% vs. 25% in patients without PSC, *p* < 0.001) and endoscopically invisible (66% vs. 21%, *p* < 0.001) [40]. Among these, crypt-cell dysplasia, hypermucinous dysplasia, and goblet-cell deficient dysplasia are the most represented and are considered higher-risk variants within the spectrum of unconventional dysplasia [97]. Hypermucinous dysplasias, for example, are associated with a *KRAS* mutation rate of 61%, significantly higher than in conventional low-grade dysplasia (4%; *p* < 0.001) or high-grade dysplasia (29%; *p* > 0.05) [98]. Similarly, high rates of *PIK3CA* (56%), *TP53* (44%), and *KRAS* (22%) mutations were reported in caliciform cell-poor dysplasia [99].

The frequency of aneuploidy is significantly higher in unconventional than in conventional dysplasias [40]. Specifically, aneuploidy was observed in 80% of hypermucinous dysplasia and 100% of crypt-cell dysplasia, rates similar to those in high-grade flat/invisible dysplasia (93%) but much higher than in low-grade conventional dysplasia (8%) or sporadic adenomas (9%) (*p* < 0.001) [100,101]. These data suggest that unconventional lesions, typical of patients with PSC-IBD, are characterized by greater genomic instability and a higher risk of progression to advanced neoplasia [40].

These findings support the current recommendations to perform annual endoscopic surveillance in patients with PSC-IBD and to consider colectomy if dysplasia is detected, given the high likelihood of progression to advanced neoplasia [40]. The identification of unconventional and invisible dysplastic features in patients with PSC-IBD could also serve as an early marker to identify patients with IBD at high risk for PSC, allowing surveillance and treatment strategies to be tailored according to the individual risk profile.

#### 5.2.4. Chemoprevention of CRC in PSC-IBD: The Role of UDCA and 5-ASA

UDCA, used in the treatment of PSC and other chronic cholestatic diseases, has been proposed as a chemopreventive agent in PSC-IBD-CRC due to its ability to modulate the biliary microenvironment in the colon and reduce the production of hydrophobic bile acids such as DCA, which are implicated in tumorigenesis [79]. However, studies on the efficacy of UDCA in the chemoprevention of CRC in PSC-IBD have provided mixed results [79,102]. A meta-analysis by Singh et al., which included eight studies with 177 cases of colorectal neoplasia among 763 patients with PSC-IBD, found that the use of UDCA was not associated with a significant reduction in the risk of colorectal neoplasia (OR 0.81); however, subgroup analysis showed that low doses of UDCA (8–15 mg/kg/day) were associated with a significant reduction in CRC risk of 81% (OR 0.19). In contrast, intermediate and high doses (15–30 mg/kg/day) did not change the risk of CRC [103]. Some studies have raised concerns about a possible increased risk of neoplasia with high doses of UDCA (>28–30 mg/kg/day), due to the concomitant increase in cytotoxic bile acid levels in the colon [67,104,105]. After OLT, the use of UDCA has not shown a positive or negative impact on the risk of colorectal neoplasia, although a large Nordic multicenter study found a significant increase in the risk of colorectal neoplasia post-OLT in patients treated with UDCA [87]. The use of 5-aminosalicylates (5-ASA) has also been investigated for a possible protective effect in PSC-IBD patients [79]. Although some studies have associated 5-ASA with a reduction in the risk of dysplasia progression (HR 0.2), others have not found a protective effect against CRC, and in a study by Jorgensen et al., the use of 5-ASA was even associated with an increased risk of colorectal neoplasia, possibly due to the suppression of antitumor activity by the anti-inflammatory effect [87,106]. In light of these data, the role of UDCA and 5-ASA in the chemoprevention of CRC in patients with PSC-IBD remains uncertain, and their routine use for this purpose is not currently recommended [67,79].

### 5.3. Other Malignancies

In patients with PSC-IBD, in addition to CCA and CRC, a significantly increased risk of other hepatobiliary and pancreatic neoplasms, including HCC, pancreatic carcinoma, and GBC, has been observed [13,67]. Table 2 provides an overview of malignancies associated with PSC and PSC-IBD, including relative risks, key risk factors, and recommended surveillance strategies.

#### 5.3.1. Hepatocellular Carcinoma

The incidence of HCC in patients with PSC remains poorly defined, but available evidence suggests a cumulative lifetime incidence of between 0.3% and 2.8% [67,107]. In a study of a population of 830 patients with PSC, 23 cases of HCC were identified, all found exclusively in patients with cirrhotic stage PSC [108]. This finding suggests that the risk of HCC in patients with PSC might be attributable primarily to the presence of cirrhosis, similar to what is observed in other chronic liver diseases. Thus, cirrhosis of the liver appears to be the major risk factor for the development of HCC in patients with PSC, whereas the risk in patients without cirrhosis is probably negligible [67,108].

An analysis conducted in patients with PSC-IBD showed a 1.8 percent incidence rate of HCC [13]. The only significant risk factor for the development of HCC was found to be age at diagnosis of PSC, with a HR of 1.33 (95% CI, 1.13–1.57; *p* < 0.001) [13]. This finding strengthens the hypothesis that advanced age is an important risk factor for HCC in patients with PSC-IBD, regardless of the presence of other comorbidities [13].

HCC is relatively rare in patients with PSC in the absence of cirrhosis [109]. Therefore, surveillance for HCC is recommended in patients with PSC who develop cirrhosis, following principles similar to those adopted for the management of cirrhosis of other etiology [109,110].

#### 5.3.2. Pancreatic Carcinoma

A recent meta-analysis of 13 cohort studies reported a significantly increased risk of pancreatic cancer in patients with IBD compared to the general population (RR 1.79; 95% CI 1.16–2.75), with similar values for CD (RR 1.42) and UC (RR 1.50) [111].

A significant increased risk has also been observed for pancreatic carcinoma in patients with PSC-IBD [13,67]. It has been reported that pancreatic carcinoma was diagnosed in 3% of patients with PSC-IBD compared with fewer than 1% in patients with isolated IBD [13]. The occurrence of PSC was found to be associated with a more than five-fold higher risk of pancreatic carcinoma compared with IBD alone (adjusted HR, 5.26; 95% CI, 2.81–9.84; *p* < 0.001) [13]. Again, age at diagnosis of PSC emerged as a significant risk factor for pancreatic carcinoma (HR, 1.76; 95% CI, 1.52–2.03; *p* < 0.001), suggesting an interaction between advanced age and tumor susceptibility in patients with PSC-IBD [13].

Despite the clear association between PSC and pancreatic cancer, specific surveillance for this malignancy is not currently recommended due to limited data availability and the lack of effective screening strategies in this population [67]. However, the strong link between PSC and pancreatic carcinoma suggests the need for further investigation to evaluate the effectiveness of any targeted surveillance strategies.

#### 5.3.3. Gallbladder Carcinoma

A high incidence of gallbladder abnormalities, including stones, cholecystitis, polyps, and GBC, has been found in patients with PSC. Importantly, while in the general population most gallbladder polyps are benign in nature, a higher incidence of dysplastic or malignant polyps has been observed in patients with PSC [13,67].

Regarding GBC, it has been reported that among patients with PSC-IBD, the risk of developing GBC is significantly higher than among patients with IBD alone (adjusted HR, 9.19; 95% CI, 2.91–29.05) [13]. Age at diagnosis of PSC was found to be the main risk factor for the development of GBC (HR, 1.44; 95% CI, 1.18–1.75; *p* > 0.001), suggesting a determinant role of age in predisposition to the development of this neoplasm [13].

Among patients with PSC-IBD who underwent cholecystectomy, about 6% developed GBC [13]. The probability of cholecystectomy was higher in patients with PSC than in those with IBD alone (HR, 1.16; 95% CI, 1.10–1.23; *p* < 0.001). Female sex was identified as an additional risk factor for cholecystectomy (HR, 1.37; 95% CI, 1.11–1.71; *p* = 0.004), suggesting a possible hormonal influence in the risk of gallbladder disease in patients with PSC [13].

According to ACG guidelines, the management of cholecystic polyps ≤ 8 mm in patients with PSC remains controversial, even in light of the non-negligible operative risk: in patients with advanced liver disease, in fact, cholecystectomy is associated with a postoperative complication rate of up to 40% [109,112,113]. A review of the literature showed that an ultrasound cutoff of 8 mm allows detection of neoplastic lesions with a sensitivity of 96% and a specificity of 53% [109,114]. Based on these data, ACG guidelines recommend ultrasound monitoring every 6 months for polyps ≤ 8 mm [109]. For polyps > 8 mm, the decision between cholecystectomy and surveillance should be based on assessment of liver function and risk of perioperative complications, such as hepatic decompensation or biliary infections [109]. Patients with advanced disease should be referred to specialized centers, preferably with liver transplantation programs [109].

In summary, patients with PSC-IBD have an increased risk not only of CCA and CRC but also of HCC, pancreatic carcinoma, and GBC. The risk of HCC is closely related to the presence of cirrhosis, while the risk of pancreatic and GBC is significantly increased in patients diagnosed with PSC compared with patients with IBD alone. Clinical management of these neoplasms requires a multidisciplinary and individualized approach, with emphasis on regular surveillance and early surgical evaluation for gallbladder neoplasms. Implementation of targeted surveillance strategies for pancreatic carcinoma could be an area of future research of clinical relevance.

## 6. Surveillance Strategies

In patients with PSC-IBD, oncologic surveillance is particularly challenging due to a significantly elevated risk of both CRC and CCA compared to IBD alone. This heightened risk underscores the necessity for structured and disease-specific surveillance protocols aimed at early malignancy detection. CRC surveillance is based on annual colonoscopy, with enhanced imaging modalities such as dye-based (DCE) and virtual chromoendoscopy (VCE) outperforming high-definition white-light endoscopy (HD-WLE) in dysplasia detection [109,115]. Advanced technologies like confocal laser endomicroscopy (CLE) and artificial intelligence (AI)-assisted analysis hold promise for further improving diagnostic yield [116,117].

CCA surveillance typically includes serial magnetic resonance imaging with cholangiopancreatography (MRI/MRCP) and serum CA 19-9 measurement, though both methods are limited by suboptimal sensitivity and specificity [109,115]. Adjunctive techniques such as biliary endoscopy with cytology and fluorescence in situ hybridization (FISH) enhance diagnostic accuracy but carry procedural risks [118,119]. Emerging tools, including AI-based risk models and bile acid profiling, may further refine surveillance strategies and enable improved risk stratification [120]. Table 3 elucidates risk groups and surveillance strategies in patients affected by PSC and PSC-IBD.

### 6.1. Surveillance for Colorectal Cancer

Guidelines from leading scientific societies, including the European Society of Gastrointestinal Endoscopy (ESGE), the European Crohn’s and Colitis Organization (ECCO), and the American College of Gastroenterology (ACG), recommend the initiation of endoscopic surveillance at the time of diagnosis of PSC, with subsequent annual follow-ups [109,115]. According to the ACG guidelines, in PSC patients younger than 15 years of age, the risk of colorectal cancer is considered to be very low; therefore, endoscopic surveillance is recommended from the age of 15 years [109,121].

ECCO also recommends annual surveillance in patients with an ileal pouch with a history of dysplasia, colorectal carcinoma, a family history of CRC, or PSC [115]. Currently, there are insufficient data, in terms of person-years of follow-up, to determine whether PSC is an independent risk factor for the development of pouch carcinoma [115]. Although the absolute number of cases of pouch neoplasia is low, annual endoscopic surveillance is recommended in patients with PSC, regardless of the presence of dysplasia in the colectomy specimen [115]. This recommendation is based on the high overall incidence of CRC in patients with PSC and the higher frequency of pouchitis in patients with PSC, which could mask early symptoms of neoplasia [90,115]. Pouchoscopies should be performed by endoscopists experienced in IBD [115,122]. It is essential that each endoscopic examination documents the following in detail: the pre-pouch ileum, the pouch body, and the ileo-pouch anal anastomosis with systematic biopsies taken from each of these sites [115,122].

Endoscopic surveillance has been associated with reduced CRC-related mortality in patients with PSC-IBD [82].

HD-WLC is considered the standard method for CRC surveillance in patients with PSC-IBD [117]. Studies have shown that HD-WLC is associated with a 28.8% detection rate of adenomas compared with 24.3% for standard definition WLC (*p* = 0.012) and a 42.2% detection rate of polyps compared with 37.8% (*p* = 0.026) [123]. However, this technique has limitations in detecting flat lesions, especially in the presence of active inflammation [123].

DCE, using carmine indigo or methylene blue, has demonstrated superior detection compared with HD-WLC [117]. A meta-analysis showed a 7% increase in detection rate over standard WLC (95% CI: 3.2–11.3), and another meta-analysis of 978 patients showed the superiority of DCE over HD-WLC with an OR of 1.94 (95% CI: 1.21–3.11; *p* = 0.006) [124,125]. DCE, however, requires a longer procedure time and optimal bowel preparation, and it is particularly useful in patients at high risk of CRC [117].

VCE, such as narrow band imaging (NBI), has also shown comparable dysplasia detection rates to DCE and HD-WLC, with the advantage of a shorter procedure time [126]. Among the new techniques, confocal laser endomicroscopy (CLE) allows real-time histologic examination and has been shown to increase detection of neoplasia 4.75-fold compared with conventional colonoscopy [116]. Autofluorescence imaging (AFI) improves the detection of dysplasia in patients with long-standing ulcerative colitis, with 100% sensitivity and a significant reduction in the need for nontargeted biopsies [127].

Randomized biopsies, taken from four quadrants every 10 cm, were historically part of standard practice during surveillance, but their efficacy is controversial [40,117]. One study reported a dysplasia detection rate of 0.2% in 1010 patients, while another showed a significantly higher detection rate of 18.1% in 442 patients with IBD, with 11.8% of dysplasia cases identified exclusively by randomized biopsies [127,128]. In patients with PSC-IBD, the detection of dysplasia by randomized biopsies was significantly more frequent than in IBD alone (15% vs. 5%; *p* = 0.003) [129]. In a retrospective study of 71 patients with PSC-UC, 46% of dysplasia cases were identified exclusively by random biopsies [129]. Furthermore, Shah et al. reported that 38% of low-grade dysplasia in patients with PSC-IBD was detected only by random biopsies, compared with 22% in patients with IBD alone (*p* = 0.01) [130]. This justifies maintaining this practice in patients with PSC-IBD, especially in the presence of a history of dysplasia or the high risk of CRC [40].

AI represents a new horizon in CRC surveillance in patients with PSC-IBD [117]. The IBD-CADe model, developed using 1266 HD-WLC images and 426 dye chromoscopy images, showed a sensitivity of 95.1%, specificity of 98.8% and accuracy of 96.8% [131]. Integration of AI models with histological and endoscopic data could optimize monitoring and improve predictive capabilities in the early detection of dysplasia and neoplasia. CLE and VCE, supported by AI algorithms, could offer clinical benefits in high-risk CRC patients by tailoring surveillance strategies according to individual risk profiles [117].

In summary, CRC surveillance in patients with PSC-IBD requires a multidimensional approach, integrating advanced imaging techniques, targeted and randomized biopsies, and the support of AI-based tools. While HD-WLC remains the standard method, DCE and VCE offer complementary advantages in high-risk patients. The integration of CLE, AFI, and AI could further improve early detection and management of dysplasia, contributing to more effective and personalized surveillance strategies in patients with PSC-IBD. Table 4 summarizes the current endoscopic surveillance strategies recommended for patients with PSC-IBD, including advanced techniques and special considerations for high-risk subgroups.

### 6.2. Surveillance for Cholangiocarcinoma

Surveillance for CCA in patients with PSC-IBD is a complex clinical challenge because of the high risk of developing CCA and the difficulty of early diagnosis.

Surveillance strategies for CCA in patients with PSC-IBD rely primarily on imaging and serum markers [67,82]. Although there are no specific prospective studies on CCA surveillance in patients with PSC, available data support the effectiveness of surveillance [115]. A registry study conducted in the United Kingdom of 284,560 patients (including 2588 with PSC) showed a significant reduction in the risk of hepatopancreatobiliary tumor-related death in patients with PSC-IBD who underwent annual surveillance by imaging, compared with those who did not undergo surveillance (hazard ratio 0.43; 95% CI: 0.23–0.8) [13]. Similarly, another retrospective study showed that patients with PSC who underwent regular surveillance had a significantly higher 5-year overall survival compared with the unsupervised group (68% vs. 20%; *p* < 0.001) and a significantly lower likelihood of developing hepatobiliary carcinoma-related adverse events in the same time interval (32% vs. 75%; *p* < 0.001) [108].

In the absence of definitive prospective studies, surveillance by imaging is currently recommended every 6 to 12 months [115]. Appropriate imaging tools may include magnetic resonance imaging with MRCP or hepatic ultrasonography, depending on available resources and patient characteristics [115].

Most referral centers perform annual or biannual MRI/MRCP in patients with PSC, with a sensitivity and specificity of 89% and 75%, respectively [132]. Transabdominal ultrasound is a cheaper and well-tolerated alternative, but with lower sensitivity (57%) and higher specificity (94%) [132]. Computed tomography (CT) shows comparable sensitivity and specificity to MRI/MRCP (75% and 80%, respectively), but is not recommended for surveillance because of radiation exposure and the use of iodinated contrast agents, as well as slightly lower diagnostic performance in some studies [108].

The tumor marker CA 19-9 is widely used in clinical practice for the diagnosis of CCA, although with limitations due to low sensitivity and specificity [67,133]. With a cutoff of 20 IU/mL, sensitivity is 78% and specificity is 67% [132]. However, about one-third of patients with CA 19-9 levels above 129 IU/mL do not actually have CCA, which highlights a high false-positive rate, especially in cases of benign biliary obstruction or bacterial cholangitis [134]. Nevertheless, the combination of elevated CA 19-9 levels (>20 IU/mL) with suspicious findings on MRI/MRCP can increase the sensitivity of CCA detection to nearly 100%, albeit with a reduction in specificity to 38% [132]. The combination of CA 19-9 and ultrasound can further improve sensitivity to 91%, with a specificity of 67% [132].

ERCP is an advanced diagnostic modality for surveillance of CCA in patients with PSC [67]. ERCP allows direct biliary sampling for cytology and other molecular analysis. The sensitivity of cytology for the diagnosis of CCA in patients with PSC is relatively low (43%), but the use of fluorescence in situ hybridization (FISH) can significantly improve it [118,119]. The combined sensitivity of ERCP and FISH has been reported to be 68%, with a specificity of 70% [82]. However, ERCP is associated with a non-negligible complication rate, including pancreatitis and cholangitis, with a hospitalization rate of more than 10% [135]. For this reason, ERCP is generally reserved for cases in which imaging and/or CA 19-9 results are positive or indeterminate [67].

Endoscopic ultrasound (EUS) with needle aspiration (FNA) has been shown to be useful in the diagnosis of CCA, particularly in cases with negative brushing cytology and no detectable masses on cross-sectional images [136]. However, similar to transcutaneous biopsies, concerns remain about the risk of tumor dissemination along the needle pathway [50]. For this reason, the use of EUS with tissue sampling varies depending on technological availability and local experience [137].

Peroral cholangioscopy, using thin, flexible endoscopes that allow direct visualization of extrahepatic biliary tract stenosis, has improved diagnostic accuracy in sporadic CCA, although specific data in patients with PSC are limited. In a study of 47 patients with PSC, the use of single-operator cholangioscopy (e.g., SpyGlass™, Boston Scientific, Marlborough, MA, USA) allowed targeted biopsies from otherwise inaccessible stenoses, but showed limited sensitivity (33%) while maintaining 100% specificity [137,138].

Given the limitations of available evidence and the variability of access to EUS and second-generation cholangioscopy, ERCP with biliary brushing currently remains the standard method for acquiring tissue samples in PSC patients with suspected CCA [118,139].

The integration of new predictive tools based on AI could further improve surveillance and early diagnosis of CCA in patients with PSC [120]. Recent studies have identified the duration of IBD and PSC, bilirubin levels, and CA 19-9 as major risk factors for developing CCA. Customized risk models, constructed using Cox proportional hazards (CoxPH), Random Survival Forest (RSF), and Gradient Boosting Survival Analysis (GBSA) algorithms, have shown superior predictive ability compared to traditional risk scores (the Mayo PSC Risk Score, MELD, and PREsTo) [120]. The CoxPH model proved particularly effective in predicting CCA risk based on clinical and biochemical variables, while RSF and GBSA exhibited signs of overfitting, meaning they performed well on the training data but lacked reliability when applied to new data, likely due to the limited sample size. This highlights the need for studies on larger cohorts to improve the generalizability of results [120]. Bile acid analysis could make an additional contribution: in models combined with clinical and laboratory variables, bile acid increased predictive ability (C-index of 0.67), suggesting that the metabolic changes associated with CCA progression could be reflected in bile acid profiles [120].

Finally, the integration of advanced imaging data (such as MRI/MRCP) with AI-based predictive models could improve the ability to identify CCA risk early in patients with PSC-IBD. The use of artificial intelligence to analyze imaging data and changes in serum biomarkers could enable more accurate risk stratification and personalization of surveillance strategies [120].

However, surveillance strategies should be implemented with caution, considering the limited availability of diagnostic tools and the possibility of false positives. In addition, treatment options for CCA remain limited: curative surgical resection is indicated only in a minority of patients, and liver transplantation is reserved for selected patients with hilar CCA in highly specialized centers [67].

In summary, surveillance of CCA in patients with PSC-IBD requires an integrated and personalized approach combining advanced imaging techniques, serum biomarkers, artificial intelligence-based predictive models, and direct biliary sampling. The evolution of predictive and diagnostic technologies could significantly improve the early detection of CCA and optimize the management of this complication in patients with PSC.

## 7. Conclusions

Oncologic risk management in PSC-IBD necessitates a multifaceted approach that integrates intensive surveillance, advanced diagnostics, and individualized therapeutic strategies. The markedly increased risk of both CRC and CCA in this population, compared to isolated IBD or PSC, reflects a complex interplay of chronic inflammation, dysbiosis, and dysregulated biliary metabolism. Emerging technologies such as artificial intelligence and microbiome profiling offer promising avenues for enhanced risk stratification and earlier detection.

Nonetheless, critical gaps remain, including the lack of reliable biomarkers and disease-specific therapeutic options. Future research should focus on elucidating the roles of the gut microbiota, immune dysregulation, and bile acid signaling in oncogenesis, with the goal of developing targeted, precision-based interventions for this high-risk cohort.

## Figures and Tables

**Figure 1 cancers-17-02165-f001:**
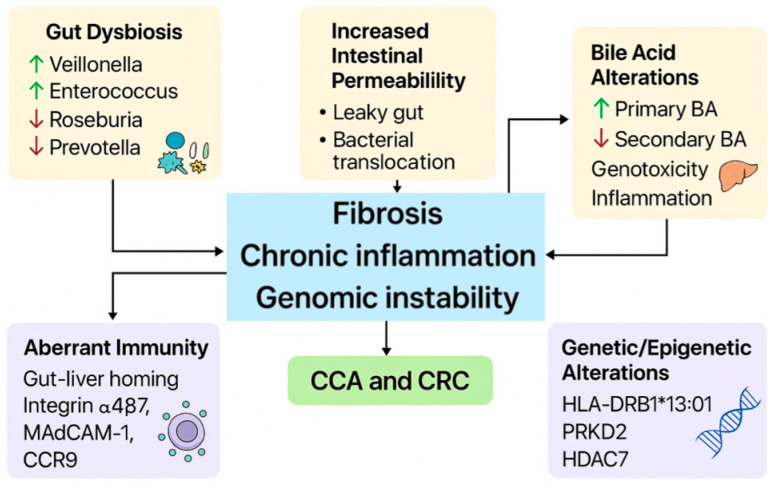
Pathogenic mechanisms linking PSC and IBD.

**Table 1 cancers-17-02165-t001:** Comparative molecular profile of PSC-IBD-associated and sporadic CRC.

Feature	PSC-IBD Associated CRC	Sporadic CRC
Onset age	Younger	Older
Tumor location	Right colon predominance	Left-sided predominance
Dysplasia type	Invisible, flat, multifocal	Visible, polypoid
Molecular alterations	p53 early mutation, low APC/KRAS	APC/KRAS mutations common
Microsatellite instability (MSI)	Rare	Present in Lynch/MSI-H sporadic
CpG island methylation (CIMP)	Less common	More common in serrated pathway

PSC, primary sclerosing cholangitis; IBD, inflammatory bowel disease; CRC, colorectal cancer; MSI, microsatellite instability; CIMP, CpG island methylator phenotype.

**Table 2 cancers-17-02165-t002:** Tumor risk and surveillance strategies in PSC and PSC-IBD.

Tumor Type	Relative Risk Increase	Risk Factors	Guideline Surveillance
Colorectal Cancer (CRC)	4–10× in PSC-IBD vs. IBD alone	Colonic involvement, duration of disease, male sex, backwash ileitis	Annual colonoscopy from IBD diagnosis (ECCO, BSG, EASL)
Cholangiocarcinoma (CCA)	~160× vs. general population	Older age, dominant strictures, elevated bilirubin, concurrent IBD	MRI/MRCP every 6–12 months ± CA 19-9 (BSG, EASL)
Gallbladder Cancer (GBC)	Increased; particularly with polyps > 8 mm	Gallbladder polyps, chronic inflammation	Ultrasound every 6–12 months (EASL)
Hepatocellular Carcinoma (HCC)	Increased only in cirrhotic PSC	Advanced fibrosis, cirrhosis	Ultrasound every 6 months in cirrhotic patients (EASL)
Pancreatic Cancer	>5× in PSC-IBD vs. IBD alone	Older age, chronic inflammation	Not routinely recommended

PSC, primary sclerosing cholangitis; IBD, inflammatory bowel disease; CRC, colorectal cancer; CCA, cholangiocarcinoma; GBC, gallbladder cancer; HCC, hepatocellular carcinoma; MRI, magnetic resonance imaging; MRCP, magnetic resonance cholangiopancreatography; CA 19-9, carbohydrate antigen 19-9; ECCO, European Crohn’s and Colitis Organisation; BSG, British Society of Gastroenterology; EASL, European Association for the Study of the Liver.

**Table 3 cancers-17-02165-t003:** Risk groups and surveillance strategies in patients affected by PSC and PSC-IBD.

Cancer Type	Risk Group	Surveillance Modality	Frequency	Notes
Colorectal Cancer (CRC)	PSC with inflammatory bowel disease (IBD)	Colonoscopy with biopsies	Annually starting at IBD diagnosis	Increased CRC risk with PSC-IBD; earlier and more frequent screening recommended
Cholangiocarcinoma (CCA)	All PSC patients	MRCP with CA 19-9	Annually	May help detect CCA early, although evidence regarding survival benefit remains inconclusive

PSC, primary sclerosing cholangitis; IBD, inflammatory bowel disease; MRCP, magnetic resonance cholangiopancreatography; CA 19-9, carbohydrate antigen 19-9.

**Table 4 cancers-17-02165-t004:** Colorectal cancer surveillance in PSC-IBD: techniques and timing.

Category	Surveillance Strategy	Frequency	Notes
Patients with PSC-IBD (UC or CD with colonic involvement)	High-definition colonoscopy (HD-WLC) with random biopsies or chromoendoscopy (DCE/VCE)	Annually, starting at IBD diagnosis	PSC-IBD has higher CRC risk than IBD alone; invisible dysplasia is frequent; chromoendoscopy improves detection (~7% more than HD-WLC)
Advanced endoscopy techniques	DCE with indigo carmine/methylene blue; NBI/VCE; CLE; AFI	Based on patient risk and lesion suspicion	DCE preferred for high-risk patients; NBI/VCE have similar sensitivity with faster execution; CLE increases detection (×4.75); AFI sensitive for early dysplasia
Artificial Intelligence (AI)	AI-assisted image analysis (e.g., IBD-CADe)	In development for future protocols	May improve dysplasia detection (95.1% sensitivity, 98.8% specificity); promising adjunct for endoscopic surveillance
Surveillance in IPAA (post-colectomy)	Pouchoscopy with systematic biopsies (pre-pouch ileum, pouch body, anastomosis)	Annually, even if no prior dysplasia	PSC-IBD patients have increased risk of pouch neoplasia and pouchitis; systematic biopsies essential even without prior dysplasia
Patients with confirmed dysplasia	Intensified surveillance or colectomy	Based on dysplasia grade (LGD vs. HGD)	PSC-IBD dysplasia (esp. invisible) has high CRC progression risk; aneuploidy in >80% of HGD cases

PSC, primary sclerosing cholangitis; IBD, inflammatory bowel disease; UC, ulcerative colitis; CD, Crohn’s disease; HD-WLC, high-definition white light colonoscopy; DCE, dye-based chromoendoscopy; VCE, virtual chromoendoscopy; NBI, narrow band imaging; CLE, confocal laser endomicroscopy; AFI, autofluorescence imaging; AI, artificial intelligence; CRC, colorectal cancer; IPAA, ileo-pouch anal anastomosis; LGD, low-grade dysplasia; HGD, high-grade dysplasia.

## Data Availability

No new data were created or analyzed in this study. Data sharing is not applicable to this article.

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
