# Peer review of "Navigating Neoplasm Risk in Inflammatory Bowel Disease and Primary Sclerosing Cholangitis"

_cancers, 2025, doi:10.3390/cancers17132165_

Round 1
Reviewer 1 Report
Comments and Suggestions for Authors
This review manuscript summarizes the mechanisms underlying high cancer risk in patients with inflammatory bowel disease (IBD)associated with primary sclerosing cholangitis (PSC). Authors of this manuscript searched for related publications published before March of 2025 through refining different combinations of search strategies using different words. This manuscript may provide detailed insights into why PSDC-IBD causes high risk of cancer development, what tools are emerging for diagnosis, and what strategies can be used for preventions of cancers such as cholangiocarcinoma (CCA) and colorectal cancer (CRC). Such review is necessary for advancing our understand of the underlying mechanisms behind high cancer risk. This review was written very comprehensively. It can make a difference in this related field. I have several comments that need to be addressed before it can be accepted for publication as below.
Major comments
- As an important review manuscript, including all important recent publications matters much. Although it is difficult to include every report related to the focus of this manuscript, more search strategies may help locate more precise research papers in PubMed. Based on the methods described in current version, words used for including papers to understand underlying molecular explanations are “molecular mechanisms”. Many other words such as molecular basis, molecular identity, regulators, mediators, and signaling molecules etc. can actually be useful for screening related papers. Do authors try more such words during the search process? More important results may be extracted to strengthen this manuscript if more words are used.
- The figure is important for review because it can really help readers understand the manuscript. One figure is included in this review manuscript, but first the resolution quality is poor. Second, it will be more helpful if important references are included or indicated in the figure. Third, more detailed information should be showed in this figure because readers can expect to get most of core findings in one review manuscript without reading through the text.
Minor comments
- Do authors have better version of the manuscript title? Here “Navigating” is used. It seems that the theme of this review cannot be completely reflected by this word.
- line 254, “International” should be “international”.
Author Response
Response to the editor(s) of ‘Cancers’ Journal
Dear Editor(s),
We sincerely thank you for giving us the opportunity to consider for re-submission and potential publication a revised version of our original manuscript entitled “Navigating Neoplasm Risk in Inflammatory Bowel Disease and Primary Sclerosing Cholangitis” by Pitoni et al.
We kindly thank the Reviewers for the precious comments. We are pleased to know that you appreciated the topic of our manuscript. The manuscript has been significantly revised and improved according to the received suggestions. Included below you can find a point-by-point response to the remarks.
Sincerely,
Alessandro Armuzzi, Professor
alessandro.armuzzi@hunimed.eu
Reviewer 1
This review manuscript summarizes the mechanisms underlying high cancer risk in patients with inflammatory bowel disease (IBD) associated with primary sclerosing cholangitis (PSC). Authors of this manuscript searched for related publications published before March of 2025 through refining different combinations of search strategies using different words. This manuscript may provide detailed insights into why PSDC-IBD causes high risk of cancer development, what tools are emerging for diagnosis, and what strategies can be used for preventions of cancers such as cholangiocarcinoma (CCA) and colorectal cancer (CRC). Such review is necessary for advancing our understand of the underlying mechanisms behind high cancer risk. This review was written very comprehensively. It can make a difference in this related field. I have several comments that need to be addressed before it can be accepted for publication as below.
Re: Thank You for Your comments, included below you can find a point-by-point response to Your remarks.
Major comments
1. As an important review manuscript, including all important recent publications matters much. Although it is difficult to include every report related to the focus of this manuscript, more search strategies may help locate more precise research papers in PubMed. Based on the methods described in current version, words used for including papers to understand underlying molecular explanations are “molecular mechanisms”. Many other words such as molecular basis, molecular identity, regulators, mediators, and signaling molecules etc. can actually be useful for screening related papers. Do authors try more such words during the search process? More important results may be extracted to strengthen this manuscript if more words are used.
Re: Thank You for this valuable suggestion. We expanded our search strategy by including the additional terms You proposed (e.g., “molecular basis”, “molecular identity”, “regulators”, “mediators”, and “signaling molecules”). However, the majority of results retrieved with these terms referred to studies that were already included in our review. In writing the pathophysiology section, we aimed to provide a broad and structured overview of the key mechanisms involved, ensuring a level of detail that is both informative and accessible. A more in-depth molecular focus would exceed the intended scope of this review, which aims to integrate clinical, pathophysiological, and surveillance aspects in a balanced and reader-friendly manner.
2. The figure is important for review because it can really help readers understand the manuscript. One figure is included in this review manuscript, but first the resolution quality is poor. Second, it will be more helpful if important references are included or indicated in the figure. Third, more detailed information should be showed in this figure because readers can expect to get most of core findings in one review manuscript without reading through the text.
Re: Thank You for pointing this out. We have improved the figure’s resolution and increased the level of detail. In addition, we have included new tables to better summarize the core findings and surveillance strategies.
Minor comments
- Do authors have better version of the manuscript title? Here “Navigating” is used. It seems that the theme of this review cannot be completely reflected by this word.
Re: We thank the reviewer for their thoughtful suggestion regarding the manuscript title. The current title, “Navigating Neoplasm Risk in Inflammatory Bowel Disease and Primary Sclerosing Cholangitis,” was carefully chosen to reflect the clinical complexity and decision-making challenges inherent to this topic. The term “navigating” was intended to capture the multifaceted and evolving nature of risk assessment and management in these patients. Moreover, the title was agreed upon in advance with the Editors, and we believe it appropriately encapsulates the scope and aims of the review. Therefore, we would prefer to retain the current wording.
- line 254, “International” should be “international”.
Re: Thank you for pointing out the typo. We have corrected “International” to “international” on line 254.
Reviewer 2 Report
Comments and Suggestions for Authors
The authors provide a detailed review on patients with primary sclerosing cholangitis (PSC), highlighting the high prevalence of concomitant inflammatory bowel disease (IBD), the increased risk of developing malignancies such as colorectal and biliary tract cancers, and current surveillance strategies. The article comprehensively covers a wide range of relevant topics and serves as a valuable resource for clinicians involved in the care of patients with PSC and IBD.
However, I have some concerns as follows:
- This review article covers a lot of topics. Therefore, the authors need to add some Tables to help organize the information. For example, summarizing cancer risk comparisons in PSC-IBD patients—such as PSC alone vs. PSC-IBD, dysplasia vs. colorectal cancer, with vs. without liver transplantation, and cancer risk in ileal pouch—would enhance clarity and deepen the clinician’s understanding.
- Section 3 and its heading appear to be repeated; this duplication should be addressed. "3. Clinical phenotype of PSC-IBD" and "3. Pathophysiological mechanisms of PSC and impact on cancer development"
- In L435 to L438, the authors describe the association between PSC and IBD. Therefore, it would be more appropriately to place into Section 3 “Clinical phenotype of PSC-IBD,” rather than in the tumor section.
- In L638 to L655, they describe carcinogenesis in the ileal pouch of PSC patients. I think it would be more appropriate to include them in Section 4.2.1, “Epidemiology and risk factors of CRC in PSC-IBD.”
Author Response
Response to the editor(s) of ‘Cancers’ Journal
Dear Editor(s),
We sincerely thank you for giving us the opportunity to consider for re-submission and potential publication a revised version of our original manuscript entitled “Navigating Neoplasm Risk in Inflammatory Bowel Disease and Primary Sclerosing Cholangitis” by Pitoni et al.
We kindly thank the Reviewers for the precious comments. We are pleased to know that you appreciated the topic of our manuscript. The manuscript has been significantly revised and improved according to the received suggestions. Included below you can find a point-by-point response to the remarks.
Sincerely,
Alessandro Armuzzi, Professor
alessandro.armuzzi@hunimed.eu
Reviewer 2
The authors provide a detailed review on patients with primary sclerosing cholangitis (PSC), highlighting the high prevalence of concomitant inflammatory bowel disease (IBD), the increased risk of developing malignancies such as colorectal and biliary tract cancers, and current surveillance strategies. The article comprehensively covers a wide range of relevant topics and serves as a valuable resource for clinicians involved in the care of patients with PSC and IBD.
However, I have some concerns as follows:
Re: Thank You for Your comments, included below you can find a point-by-point response to Your remarks.
- This review article covers a lot of topics. Therefore, the authors need to add some Tables to help organize the information. For example, summarizing cancer risk comparisons in PSC-IBD patients—such as PSC alone vs. PSC-IBD, dysplasia vs. colorectal cancer, with vs. without liver transplantation, and cancer risk in ileal pouch—would enhance clarity and deepen the clinician’s understanding.
Re: Thank You for Your thoughtful suggestion. In response, we have added several summary tables to improve clarity and enhance the reader’s understanding of key oncological risks in PSC-IBD
- Section 3 and its heading appear to be repeated; this duplication should be addressed. "3. Clinical phenotype of PSC-IBD" and "3. Pathophysiological mechanisms of PSC and impact on cancer development"
Re: Thank You for pointing this out. The duplication in section headings has been corrected. The structure has been revised to avoid overlap and ensure logical flow.
- In L435 to L438, the authors describe the association between PSC and IBD. Therefore, it would be more appropriately to place into Section 3 “Clinical phenotype of PSC-IBD,” rather than in the tumor section.
Re: Thank You for this thoughtful observation. We fully understand that the content in lines 435–438 could be considered more appropriate for the section on the clinical phenotype of PSC-IBD. However, we intentionally placed it within the subsection on colorectal cancer epidemiology, as it specifically refers to the increased incidence of CRC in this subgroup. In designing the structure of the manuscript, we aimed to organize the content thematically and avoid unnecessary repetition, ensuring a clear and coherent presentation. While Your suggestion is certainly valid, we believe the current placement supports a more streamlined and logical flow of the topics discussed.
- In L638 to L655, they describe carcinogenesis in the ileal pouch of PSC patients. I think it would be more appropriate to include them in Section 4.2.1, “Epidemiology and risk factors of CRC in PSC-IBD.”
Re: Thank You for the suggestion. We agree that the paragraph discussing carcinogenesis in the ileal pouch (L638–L655) aligns better with Section 4.2.1 (as part of the structural revision of the manuscript, section 4.2.1 has been renumbered as 5.2.1). We have relocated it accordingly to improve coherence.
Reviewer 3 Report
Comments and Suggestions for Authors
This is a comprehensive, well-written, well-structured review on IBD-PSC and is timely. Thank you for your hard work, I learned a lot by reading the manuscript. I have a few suggestions and comments to improve readability.
Line 35: there is no mention of microRNA and early CCA detection in the manuscript. Unless the authors are willing to add pertinent paragraph in the main body, please remove the statement from the abstract.
Line 56-58: the sentence “Current…” is redundant, this is already stated in line 50-54, please remove it.
Line 97, 100, 195: please change “colic” to “colonic”, “colic” means abdominal pain
Line 108-109: please rephrase, did you mean IBD in the setting of concomitant PSC may not be as clinically quiescent as previously believed?
In Figure 1, please add footnotes, especially for “TLR”; this is not mentioned in the main text
Line 174, could you change “increased abundance” to “excess” please?
Line 414-416: in the study [13], did IBD precede PSC in all patients? If not, the statement may mislead regarding the sequence of events.
Line 430-432: Is there any hypothesis/pathogenesis regarding increased risk of CRC in PSC patients without IBD? Could you add it if available, please?
Line 442-444: please clarify and rephrase, did you mean after OLT and colectomy, an increased risk of dysplasia and pouch carcinoma was observed in patients with PSC-IBD, compared with patients without PSC who had colectomy (and IPAA) and OLT? If this is the case, what was the indication for OLT in patients without PSC?
Line 481: It seems “Histological characteristics of dysplasia associated with PSC-IBD” would be appropriate, based on the contents of the section.
Line 491: Crypt cell dysplasia, hypermucinous and goblet-cell deficient dysplasia are higher-risk forms amongst unconventional dysplasia, could you rephrase it and clarify it, please?
Line 491, 495: please change “cryptic dysplasia” to “crypt cell dysplasia” and “caliciform cell poor dysplasia” to “goblet-cell deficient dysplasia”.
Line 540-557: Is there a study directly comparing HCC risk in PSC vs. PSC-IBD?
Line 558-571: Is there a study comparing pancreatic cancer risk in isolated IBD patients vs. general population? This would be pertinent for the theme of the review.
Line 572-606: Is there a study directly comparing gallbladder cancer risk in isolated IBD patients vs. general population?
Line 644: please clarify, did you mean “including a meta-analysis of 11 studies published in 2021”?
Line 649: please change “colectomy surgical piece” to “ colectomy specimen”
Line 666: please change “while a” to “and another”
Line 668, 671” please change “performance time” and “execution time” to “procedure time”
Line 748: did you mean cross-sectional images?
Line 760: please rephrase and clarify, did you mean “… tissue samples in PSC patients with suspected CCA”?
Line 770: Could you explain what “overfitting problems” mean, please?
I don’t think you need “Discussion” section for a review manuscript. This section seems to be a lengthy summary of your main body which is well-presented and detailed already. The journal does not require discussion section for a review paper either. I recommend you remove the discussion section entirely.
There are mistakes in the numbering of the headings and subheadings, there are two #3. “Pathophysiological mechanisms of PSC and impact on cancer development” should be #4, and subsequent numberings need to be updated.
Line 316-330: spacing for this paragraph is different from the rest
There are minor typographical errors: i.e., after the first use of “CCA” for cholangiocarcinoma, please use “CCA” throughout the manuscript. Please use uniform spelling: CA 19-9 is spelled as Ca 19-9 in the abstract, for example.
Author Response
Response to the editor(s) of ‘Cancers’ Journal
Dear Editor(s),
We sincerely thank you for giving us the opportunity to consider for re-submission and potential publication a revised version of our original manuscript entitled “Navigating Neoplasm Risk in Inflammatory Bowel Disease and Primary Sclerosing Cholangitis” by Pitoni et al.
We kindly thank the Reviewers for the precious comments. We are pleased to know that you appreciated the topic of our manuscript. The manuscript has been significantly revised and improved according to the received suggestions. Included below you can find a point-by-point response to the remarks.
Sincerely,
Alessandro Armuzzi, Professor
alessandro.armuzzi@hunimed.eu
Reviewer 3
This is a comprehensive, well-written, well-structured review on IBD-PSC and is timely. Thank you for your hard work, I learned a lot by reading the manuscript. I have a few suggestions and comments to improve readability.
Re: Thank You for Your comments, included below you can find a point-by-point response to Your remarks.
- Line 35: there is no mention of microRNA and early CCA detection in the manuscript. Unless the authors are willing to add pertinent paragraph in the main body, please remove the statement from the abstract.
Re: Thank You for Your comment. We have removed the reference to microRNA and early CCA detection from the abstract, as this topic is not discussed in the main text.
- Line 56-58: the sentence “Current…” is redundant, this is already stated in line 50-54, please remove it.
Re: Thank You for Your suggestion. We have removed the redundant sentence to avoid repetition with the previous lines.
- Line 97, 100, 195: please change “colic” to “colonic”, “colic” means abdominal pain
Re: Thank You for pointing this out. We have corrected “colic” to “colonic” in all relevant instances.
- Line 108-109: please rephrase, did you mean IBD in the setting of concomitant PSC may not be as clinically quiescent as previously believed?
Re: Thank You for the clarification. We have rephrased the sentence to state that IBD, in the setting of concomitant PSC, may not be as clinically quiescent as previously believed.
- In Figure 1, please add footnotes, especially for “TLR”; this is not mentioned in the main text
Re: Thank You for Your comment. We have removed the original version of Figure 1 and replaced it with a corrected version.
- Line 174, could you change “increased abundance” to “excess” please?
Re: Thank You for Your suggestion. We have replaced “increased abundance” with “excess” for greater clarity.
- Line 414-416: in the study [13], did IBD precede PSC in all patients? If not, the statement may mislead regarding the sequence of events.
Re: Thank You for Your helpful observation. We have clarified in the manuscript that the temporal relationship between IBD and PSC was not consistent across patients in study [13]. In particular, CRC developed in some cases after IBD but before PSC diagnosis, while in others it occurred shortly after PSC onset. We have also highlighted that a uniform disease chronology should not be assumed in this cohort.
- Line 430-432: Is there any hypothesis/pathogenesis regarding increased risk of CRC in PSC patients without IBD? Could you add it if available, please?
Re: Thank You for Your insightful comment. We have added a brief paragraph discussing the hypothesized role of chronic bile acid exposure—particularly secondary bile acids—in inducing DNA damage and promoting a pro-neoplastic microenvironment in patients with isolated PSC. This may help explain the increased CRC risk even in the absence of concomitant IBD.
- Line 442-444: please clarify and rephrase, did you mean after OLT and colectomy, an increased risk of dysplasia and pouch carcinoma was observed in patients with PSC-IBD, compared with patients without PSC who had colectomy (and IPAA) and OLT? If this is the case, what was the indication for OLT in patients without PSC?
Re: Thank You for this important clarification request. The comparison refers to patients with PSC-IBD undergoing both colectomy and liver transplantation, versus patients with IBD without PSC who underwent colectomy alone. The increased risk of pouch malignancy has been reported primarily in the context of PSC-IBD after OLT, and no specific indication for OLT was assumed in the non-PSC group.
- Line 481: It seems “Histological characteristics of dysplasia associated with PSC-IBD” would be appropriate, based on the contents of the section.
Re: Thank You for Your suggestion. We have updated the section title to “Histological characteristics of dysplasia associated with PSC-IBD” to better reflect the content.
- Line 491: Crypt cell dysplasia, hypermucinous and goblet-cell deficient dysplasia are higher-risk forms amongst unconventional dysplasia, could you rephrase it and clarify it, please?
Re: Thank You for the helpful comment. We have rephrased the sentence to clarify that the most common and higher-risk forms of unconventional dysplasia in PSC-IBD include crypt cell dysplasia, hypermucinous dysplasia, and goblet-cell deficient dysplasia.
- Line 491, 495: please change “cryptic dysplasia” to “crypt cell dysplasia” and “caliciform cell poor dysplasia” to “goblet-cell deficient dysplasia”.
Re: Thank You for the correction. We have replaced “cryptic dysplasia” with “crypt cell dysplasia” and “caliciform cell poor dysplasia” with “goblet-cell deficient dysplasia” as suggested.
- Line 540-557: Is there a study directly comparing HCC risk in PSC vs. PSC-IBD?
Re: Thank You for Your comment. To our knowledge, no studies have directly compared the risk of hepatocellular carcinoma (HCC) between patients with PSC and those with PSC-IBD. We have therefore clarified in the text that the incidence of HCC in PSC remains poorly defined and is primarily associated with the development of cirrhosis, regardless of IBD status. Available data suggest that HCC is rare in non-cirrhotic PSC patients and that the presence of cirrhosis and older age are the main risk factors for HCC in this population.
- Line 558-571: Is there a study comparing pancreatic cancer risk in isolated IBD patients vs. general population? This would be pertinent for the theme of the review.
Re: Thank You for Your comment. We have added that, while no studies have specifically compared pancreatic cancer risk in isolated IBD patients vs. the general population, recent meta-analyses suggest a significantly increased risk, supporting the relevance of this topic.
- Line 572-606: Is there a study directly comparing gallbladder cancer risk in isolated IBD patients vs. general population?
Re: Thank You for Your important question. To our knowledge, there are currently no studies directly comparing the risk of gallbladder cancer (GBC) in patients with isolated IBD versus the general population. The available literature mainly focuses on patients with concurrent PSC, in whom the risk of GBC is significantly elevated. Furthermore, existing guidelines do not recommend routine gallbladder surveillance in IBD patients without PSC.
- Line 644: please clarify, did you mean “including a meta-analysis of 11 studies published in 2021”?
Re: Thank You for Your observation. We have clarified that the sentence refers to a meta-analysis of 11 studies published in 2021.
- Line 649: please change “colectomy surgical piece” to “colectomy specimen”
Re: Thank You for Your comment. We have replaced “colectomy surgical piece” with the correct term “colectomy specimen”.
- Line 666: please change “while a” to “and another”
Re: Thank You for the suggestion. We have changed “while a” to “and another” for clarity.
- Line 668, 671” please change “performance time” and “execution time” to “procedure time”
Re: Thank You for the observation. We have updated “performance time” and “execution time” to “procedure time” for consistency and clarity.
- Line 748: did you mean cross-sectional images?
Re: Thank You for Your suggestion. We have clarified that the correct term is “cross-sectional images”.
- Line 760: please rephrase and clarify, did you mean “… tissue samples in PSC patients with suspected CCA”?
Re: Thank You for pointing this out. We have revised the sentence to explicitly state that we refer to “tissue samples in PSC patients with suspected CCA”.
- Line 770: Could you explain what “overfitting problems” mean, please?
Re: Thank You for Your request. We clarified that “overfitting problems” refer to the tendency of more complex models, such as RSF and GBSA, to perform well on training datasets but poorly on new data due to the limited sample size, thus reducing generalizability.
- I don’t think you need “Discussion” section for a review manuscript. This section seems to be a lengthy summary of your main body which is well-presented and detailed already. The journal does not require discussion section for a review paper either. I recommend you remove the discussion section entirely.
Re: Thank You for the helpful suggestion. We agree with Your assessment and have removed the “Discussion” section in accordance with the journal’s format for review articles.
- There are mistakes in the numbering of the headings and subheadings, there are two #3. “Pathophysiological mechanisms of PSC and impact on cancer development” should be #4, and subsequent numberings need to be updated.
Re: Thank You for Your observation. We corrected the numbering of headings and subheadings throughout the manuscript to ensure consistency.
- Line 316-330: spacing for this paragraph is different from the rest
Re: Thank You for the technical note. We have adjusted the spacing in this paragraph to match the rest of the manuscript.
- There are minor typographical errors: i.e., after the first use of “CCA” for cholangiocarcinoma, please use “CCA” throughout the manuscript. Please use uniform spelling: CA 19-9 is spelled as Ca 19-9 in the abstract, for example.
Re: Thank You for the detailed review. We have corrected the abbreviation usage (e.g., “CCA” consistently after first mention) and standardized the spelling of CA 19-9 throughout the manuscript, including the abstract.
Round 2
Reviewer 1 Report
Comments and Suggestions for Authors
Concerns were addressed, and it can be accepted for publication.